# Lessons from Disability Counting in Ecuador, with a Contribution from Primary Health Care

**DOI:** 10.3390/ijerph18105103

**Published:** 2021-05-12

**Authors:** Debrouwere Inge, Álvarez Vera Pedro Celestino, Pavón Benítez Ximena del Carmen, Rosero Arboleda Celia Katherine, Prinzie Peter, Lebeer Jo

**Affiliations:** 1Fundación Tapori Paladines de la Felicidad, Quito 170308, Ecuador; palvarez@ups.edu.ec (Á.V.P.C.); coordinatapori@gmail.com (P.B.X.d.C.); krosero@hospitalvozandes.com (R.A.C.K.); 2Department of Psychology, Universidad Politécnica Salesiana—Sede Quito, Quito 170308, Ecuador; 3Teaching Department, Hospital Voz Andes Quito, Quito 170308, Ecuador; 4Dominiek Savio Institute, Dienstencentrum GID(t)S, 8830 Hooglede-Gits, Belgium; prinzie@essb.eur.nl; 5Department of Psychology, Education and Child Studies, Faculty of Social and Behavioural Sciences, Erasmus University Rotterdam, 3000 DR Rotterdam, The Netherlands; 6Disability Studies, Faculty of Medicine and Health Sciences, Family Medicine & Population Health, University of Antwerp, 2610 Wilrijk, Belgium; jo.lebeer@uantwerpen.be

**Keywords:** disability, measurement, prevalence, cross-sectional studies, longitudinal studies, needs-assessment, primary health care, self-perception

## Abstract

Disability data are essential for policy. Yet, the predominant use of disability prevalence for service planning reflects dichotomous counting, increasingly less compatible with current disability thinking. Difficulties relate to variations in rates, the lack of matching with needs, and the use of prevalence to compare disability situations. From the perspective of Primary Health Care (PHC), we explore methods for disability counting regarding the usefulness of prevalences in identifying persons with disabilities and meeting their needs with local service implementation. First, we analyze the methods and results of six national cross-sectional studies in Ecuador. Then, we present a case about an exploratory needs-driven method for disability counting in a local PHC setting. The analysis of variations in rates focuses the attention on reasons for and risks of a priori exclusion of persons with disabilities from services. Longitudinal disability counting as a collateral result of meeting needs in the PHC setting yields local disability data worthy of further exploration. Thinking about disability counting from a PHC scope in a developing country prompted reflection on the comparison of prevalences to evaluate disability situations. Findings invite further exploration of the needs-driven counting method, its contributions to planning local services, and complementarity with cross-sectional disability counting.

## 1. Introduction

Countries that ratify the Convention on the Rights of Persons with Disabilities will “undertake to collect appropriate information, including statistical and research data, to enable them to formulate and implement policies to give effect to the present Convention” [1]. Three arguments underscore the importance of disability statistics. First, data are used to visualize the impact of disability as “counting disability is a political arithmetic, used to galvanize awareness of the relationship between society and disablement” [2]. Second, disability measurement assists in formulating disability policies and implementing programs for service provision. And third, comparisons of data are used to monitor the level of functioning in the population, evaluate the results of policy implementation, and assess the equalization of opportunities [3].

Disability prevalence is one of the most frequently mentioned indicators in disability data. Since the release of the World Report on Disability in 2011, many documents begin by mentioning that about 15% of world’s population experiences disability [4]. There is no doubt that identifying and counting persons with disabilities is an essential step to match service delivery to global, national, and local disability-related need and demand. However, some questions and controversies regarding the aforementioned arguments on the importance of disability data arise when counting persons with disabilities becomes the starting point in disability statistics.

One problem concerns the wide range of disability rates. Cross-country disability rates range from 4.3% (Norway and Ireland) to 35.9% (Swaziland), implying “substantially greater variations across countries in reported disability than may be plausible” [4]. Moreover, national data from the World Report on Disability show large differences with data from other studies within the same country. The report also states that “while progress is made, accurate data on disability are mostly lacking for developing countries” [4].

A second problem is that knowing how many people have disabilities does not necessarily lead to identifying and meeting disability-related needs. Whereas disability rates are often the first and only mentioned data, it is the nature of need and demand of persons with disabilities that are paramount for policy and planning. Research shows that “the simple reporting of only the prevalence of disability fails to capture the differences between communities” and “does not highlight the needs of different age groups or across different levels of severity” [5]. Counting cases a priori by first defining in a dichotomous way who is and who is not a person with disability risks “excluding people in need of targeted interventions” [6].

A third difficulty is that using the rates for evaluation of disability situations and policy implementation can lead to confusion as evaluation implies comparison and data may be compared in different ways, for example, with previous disability rates. Yet, a lower rate is not necessarily evidence of good governance and policies. Indeed, “there may even be a risk that measuring disability prevalence is used as evidence that action is planned or taken” [7]. Further, data may be compared with data from other population groups, with national goals or international references, but this raises the question as to what the gold standard should be. Finally, comparing disability prevalences is neither helpful in evaluating policy implementation when results are evaluated from a personal perspective nor for monitoring the implementation of the Convention of Rights of Persons with Disabilities when focusing on human rights.

In past decades, the disability concept has evolved from a consequence of disease and impairment at individual level to a biopsychosocial paradigm in which “disability results from the interaction between persons with impairments and attitudinal and environmental barriers that hinders their full and effective participation in society on an equal base with others”, as stated in the Convention on the Rights of Persons with Disabilities [1]. Disability now is understood as a universal multidimensional human experience that balances on a continuum of functioning [8]. This paradigm shift inevitably affects disability counting, moving beyond a quantitative and dichotomous approach that starts with dividing population groups into separated groups: persons without disabilities who do not need disability-related services and persons with disabilities in need of services. This change in thinking is reflected in the use of the International Classification of Functioning, Disability, and Health (ICF) as an underlying data standard for identifying needs and counting persons with disabilities, and for new applications in disability measurement and statistics that combine different disability measurement tools and triangulate data [9,10,11]. This encourages reflection on the appropriateness or not of the use of disability rates as a starting point for identifying persons with disabilities and collecting information on their needs.

This study is part of an ongoing research project on strategies to improve matching health service delivery to local disability-related needs, initiated by the local primary health and community-based rehabilitation center in Quito, Ecuador. The general objective of this paper is to explore methods to identify persons with disabilities and their needs over time from the perspective of Primary Health Care (PHC), as a primordial step in defining, implementing, and evaluating community-based health services on the basis of the local population’s disability profile. The specific objectives of this paper are to (1) summarize results of Ecuadorian national surveys on disability and to reflect on the usefulness of the widely varying disability prevalences, and (2) present a longitudinal community-based method for identifying needs of persons with disabilities that mainly relies on self-perception, and to explore if and how this method can be complementary to national cross-sectional identification methods for local disability-related service implementation.

## 2. Materials and Methods

Methods and results of this paper are structured in two parts corresponding to the cross-sectional dichotomous and the longitudinal needs-driven approaches to disability counting. After briefly outlining the Ecuadorian context, the first part deals with a comparison of the methods and results of six national Ecuadorian cross-sectional disability prevalence studies, as a background to disability statistics in the country. The findings of this analysis set the scene for the second part, in which we present a case report on exploratory longitudinal disability counting in a local PHC setting in Ecuador, promoting better information systems for local disability data collection for local disability policies and service implementation.

### 2.1. Part 1: A Comparison of National Cross-Sectional Disability Statistics in Ecuador

We carried out a bibliographic review of national studies on identifying and counting persons with disabilities published by the Ecuadorian government since the local Health Center started its activities in 1994. Six cross-sectional studies were identified. We examined the official reports and final documents, retrieved from the National Council on Disabilities (Consejo Nacional de Discapacidades—CONADIS), and compared their methodology and results, focusing our archival analysis on specific topics such as the definition of disability and the purpose of counting, methods for data collection and data analyses, reasons for specific methodological choices, and study results. Comparative content analysis allowed data triangulation in order to corroborate data or to understand the reasons that explain the discovered differences [12].

### 2.2. Part 2: A Case Study of Disability Counting in a PHC Setting

Data on the way of identifying and supplying disability-related needs and demands and on the registration of data of persons with disabilities in the PHC setting were collected through long-term participatory observation and from administrative databases and clinical files from 1994 to 2020. A preliminary report on this local method of collecting disability data was presented for discussion to the organization’s board and team, which allowed for a validity check of findings by triangulation among researchers. The case report presented in this paper describes, first, the dynamic between the evolving disability concept, the longitudinal identification of persons with disabilities and their needs and demands, and the organization of supply in the health center. Then, we focus on the reliability and validity of local disability data, particularly regarding the disability rate.

## 3. Results

Since 1994, the Ecuadorian population has grown from approximately 11,200,000 people to more than 17,000,000 in 2020. In this time, the median age increased from 21 to 27.9 years and life expectancy from 70.5 to 77.2 years. The annual population growth rate is now 1.4%. The under-five mortality rate decreased from 42.9 per 1000 live births in 1994 to 14 per 1000 live births. The Ecuadorian Human Development Index value for 2019 is 0.76, which put the country in the high human development category, positioned at rank 86 out of 189 countries and territories. Yet, according to the definition of the International Monetary Fund, Ecuador still is a developing country, with a gross domestic income per capita of 6184 USD in 2019. Ecuador belongs to the quintile of the countries with the greatest inequality, with a Gini coefficient that has fluctuated between 45 and 50 over the past ten years [13]. Inequality affects above all the Afro-Ecuadorian and indigenous population groups, 7.18% and 7.06% of the population respectively, especially in rural and marginal urban areas [14].

### 3.1. Part I: Ecuadorian Disability Rates from Six Cross-Sectional Studies

In this dynamic, diverse, and divergent context, six national cross-sectional studies on disability data were carried out between 1994 and 2012. To set the scene for the analysis and comparison of methods and results, we summarize in Table 1 the purposes of the studies, how disability was defined, the age of the population groups studied, their sample size and sampling method, how information was collected, and the resulting disability rate. Each study has singularities that are worth noting before comparing the results.

#### 3.1.1. Estudio de la Situación Actual de las Personas con Discapacidad en el Ecuador—ESADE (1994–2000)

A research group from the National Council on Disability (Consejo Nacional de Discapacidades—CONADIS), the Central University of Ecuador (Universidad Central del Ecuador—UCE), and the Spanish Institute for Migrations and Social Services (Instituto de Migraciones y Servicios Sociales de España—IMSERSO) carried out the cross-sectional ESADE study titled *Current Situation of Persons with Disabilities in Ecuador*. The study yields a high disability rate, without taking into account children younger than 5 years—more than 10% of the Ecuadorian population at the time.

Consistent with international concerns about underestimation of disability rates in low-income countries, the ESADE researchers stated that due to prejudice and misconceptions of the family and the Ecuadorian society, there is a tendency to hide problems related to deficiency, disability, and handicap, which may have led to subregistration of the real prevalence of people with disabilities, as persons “may feel stigma or shame at identifying themselves as disabled” [20,21,22,23]. Yet, our analysis of the ESADE results uncovers inconsistencies which suggest that the presented disability rate might be overestimated. Whereas the WHO emphasizes that, “if the selection unit is the individual, then the individual will also be the measurement unit”, Table 2 on impairments or deficiencies that produce disability reveals an erroneous equalization of the number of impairments with the number of persons with disabilities in the ESADE study [3].

The sum of the percentages of each deficiency type is exactly 100%, equal to the entire universe of deficiencies. The sum of the absolute numbers of all these deficiencies is 1,600,000, a number equal to 13.2% of the Ecuadorian population in 1996 [24]. Yet, immediately following the table, the report indicates that “all the investigated persons presented more than one deficiency, and there was great diversity in their combinations” [20]. Thus, the disability rate must be lower than 13.2% as deficiency is not equal to disability and not all the persons with disabilities have exactly one deficiency, although the report does not indicate how many deficiencies a person with disability presented on average.

#### 3.1.2. National Censuses (2001 and 2010)

The use of an a priori disability screener based on self-reporting, as implemented in the censuses, is a common first step in formulating disability policy, which begins by searching for persons with disabilities for whom, subsequently and in an exclusive way, disability-related services will be planned. Both rates from the censuses, conducted by the Ecuadorian National Institute of Statistics and Census (Instituto Nacional de Estadística y Censos—INEC), are considerably lower than the currently estimated 15% worldwide average, giving weight to the generally accepted statement that “countries reporting a low disability prevalence rate—predominantly developing countries—tend to collect disability data through censuses” [4]. Frequently cited reasons for low rates from censuses in developing countries again relate to underreporting due to negative cultural attitudes, prejudice, and stigma [7].

#### 3.1.3. World Health Survey (2002–2004)

The World Report on Disability used two sources to estimate global disability prevalence: the World Health Survey (WHS) and the Global Burden of Disease [4]. The WHS was performed by the WHO in 70 countries. In Ecuador, interviewers applied the long version of the WHS cross-culturally applicable questionnaire on eight domains related to health and functioning, namely affect, cognition, interpersonal relationships, mobility, pain, sleep and energy, self-care, and vision. The short version, mostly applied in developed countries, included only one question per domain [25]. We found no explanation of why different questionnaires were applied in these countries.

“Possible self-reported responses to the questions on difficulties in functioning included: no difficulty, mild difficulty, moderate difficulty, severe difficulty, and extreme difficulty”, and composite disability scores were calculated to produce a continuous score range [4]. Afterward, a cutoff point had to be created in order to define how many persons from the sample made up the universe of persons with disabilities. A threshold of 40 on a scale from 0 (no functioning difficulty) to 100 (complete difficulty) was set to include everyone who experienced significant difficulties in everyday live. This cutoff point was “the average score for all WHS survey respondents who reported any degree of functioning difficulties and those reporting chronic disease” [4]. A threshold of 50 was set to estimate the prevalence of persons experiencing very significant difficulties [4].

Weighted data sets from 59 countries, including Ecuador, were used to calculate the estimated prevalence of disability in the world’s population of adults aged 18 years and older. Data from 11 developed and high-income countries were unweighted without the report giving specific reasons for this matter [4].

The World Report on Disability states that “across all 59 countries, the average prevalence rate in the adult population derived from the WHS was 15.6%” [4]. At the same cutoff point of 40, the Ecuadorian disability prevalence in the population over 18 was 13.6% [4]. The report does not mention the national prevalence estimate of persons with severe disabilities, corresponding to threshold 50. At the global level, this “average prevalence rate for adults with very significant difficulties” was estimated at 2.2% and the remaining group, approximately 13% of the world’s population, would experience mild to moderate disability [4].

#### 3.1.4. Encuesta Nacional de Discapacidades (2004)

This National Survey on Disability, inserted in the quarterly National Survey of Employment, Unemployment, and Underemployment, conducted by INEC, was conducted by nearly the same group of researchers from the National Disability Council as the research team for the ESADE study, and there are other similarities between both surveys. The total numbers of persons with disabilities, including a small number of children with disabilities younger than five years in the Encuesta Nacional de Discapacidades (END), are nearly the same. Furthermore, its estimated disability rate, mentioned in the comprehensive compilation of country-reported disability rates from censuses and surveys in the World Report on Disability, is close to the result of the ESADE study [4]. The use of ICF to complement data of the ESADE study was certainly well-intentioned in this study. Unfortunately, confusion about overlapping definitions of persons with an impairment and persons with activity limitation and participation restriction led to a similar error related to the unit of measurement.

In this case, the END report contains a table that lists persons with disabilities aged five years or older with an impairment together with persons with disabilities aged five years or older with mild to moderate activity limitation and participation restriction, and another group with severe limitation and restriction. Along with a small group of children with disabilities younger than five years, the sum of these four groups equals 1,608,334 persons, presented as the Ecuadorian universe of persons with disabilities (see Table 3).

The END report defines a person with disability as “a person who experiences some degree of activity limitation and participation restriction, originated in an impairment” [18]. This means that impairment is “conditio sine qua non” for any degree of functional limitation and that “persons with impairment” and “persons with activity limitation and participation restriction” are overlapping groups. Summing these groups inflates the disability rate incorrectly. Consequently, the rate should be lower than 12.14%, although data for specific calculations are missing.

#### 3.1.5. Misión Manuela Espejo (2009–2012)

To fulfill the commitment acquired by signing the Convention of Rights of Persons with Disabilities in 2007, the Ecuadorian government conducted an action research called the Manuela Espejo Solidarity Mission (MSME) to obtain detailed information for disability policy, as one of the first needs detected was the lack of information about persons with disabilities, in terms of their residence, health status, and socioeconomic situation [19]. In addition to identifying and georeferencing persons with disabilities in the country, biopsychosocial and clinical genetic characteristics had to be described, biomedical etiology of disability investigated, and principal needs identified [19].

As a way of self-reporting, the whole population was asked to hoist a white flag in any house in which a person with disability was living. Local authorities and social organizations were requested to assist in this identification process. At the moment, no explicit definition of “person with disability” was given [26]. People from lower socioeconomic classes did not hesitate to hoist a flag, but participation of higher socioeconomic classes was limited; priority was given to assist those without financial resources in order to provide an immediate response to the critical cases [27]. The MSME visited more than a million households.

A biopsychosocial disability paradigm was established as the conceptual framework, but the brigades, consisting of Ecuadorian general practitioners, military personnel for guidance and GPS management, and community leaders, along with Cuban specialists in clinical genetics and defectology, worked from a strong biomedical and disorder-based perspective [19]. Type and degree of disability were determined and classified, using a classifier for intellectual disability and another for physical–motor, auditory, visual, mental, organic and visceral, and mixed or multiple disability [19].

The report mentions that once a priority need was detected, persons with severe functioning limitations were provided with special medical assistance, delivery of technical aids, and eventually any other required support, such as housing. Persons with mild to moderate functioning limitations were excluded from the universe of persons with disabilities entitled to services, as well as persons with temporal disabilities or with somatic or visceral disabilities, except for chronic renal failure [19]. The report does not mention what happened to the persons who were identified with mild to moderate disabilities.

While initially about 1,000,000 persons identified themselves as a person with disability, this study ultimately registered 293,743 persons with disabilities, equivalent to a rate of 2.02% [19,26]. This is the lowest rate estimated by all the studies presented in this paper. Intellectual disability was found in 71,417 persons, resulting in a rate of 0.49% or approximately one in two hundred persons. To date, the new government continues with the project “Las Manuelas” in search for updated information [28].

#### 3.1.6. Comparison of Results

Figure 1 visualizes how national disability prevalences from the cross-sectional studies fluctuate between 2.02 to 13.6 percent. Yet, changes in rates do not show a particular pattern over time and do not allow for studying associations between prevalences and sociodemographic, political, or economic dynamics.

The high disability rates of ESADE and END are the closest to the estimated global WHO rate of 15%. Yet, the analysis of the results of these studies corroborates remarks in the World Report on Disability about poor data quality, as confusion about the measurement unit resulted in higher than the actual rates in both studies [4].

The ICF-based WHS also yields a high rate, corroborating that “countries reporting higher disability prevalence tend to collect their data through surveys and apply a measurement approach that records activity limitations and participation restrictions in addition to impairments” [4]. The MSME study, which has the lowest disability rate, confirms that “impairment-based questionnaires only identify a small proportion of people with disabilities, the visibly and severely disabled” [7]. There is no doubt that disability rates are influenced by the adopted disability paradigm.

The low disability rate in the MSME also relates to the use of a high threshold. By including only persons with severe functioning difficulties in its universe of persons with disabilities entitled to services, professional second opinion excluded three quarters of persons who had self-reported as persons with disabilities. Otherwise, the WHS used a gradient from no limitations to severe functioning difficulties in all the persons from the sample, followed by an a posteriori cutoff. This approach yields higher disability rates and reduces the risk of exclusion.

Self-reporting was also used in the national censuses, but in both studies, far fewer persons than in the MSME labeled themselves as persons with disabilities. As previously noted, respondents may deny their condition due to negative cultural attitudes. Nevertheless, stigma and shame appeared of little importance when hoisting flags in the MSME, in which a positive qualification as a person with disability entailed an immediate benefit.

The comparison of disability rates from these cross-sectional studies confirms that disability counting is strongly influenced by the researchers’ methodological decisions and choices, particularly on the purpose of measuring, the disability paradigm, the choice between the use of surveys in samples, the use of closed yes–no questions or self-reporting in general populations without prior definition of disability, and the implementation of thresholds.

### 3.2. Part 2: The Case of Tapori: A Longitudinal Approach to Disability Counting in a PHC Setting (1994–2020)

The foregoing analysis proves that choosing data from cross-sectional studies for disability service planning and evaluation in a PHC setting, which aims to address local disability-related needs and demands over time and for all its users, is not straightforward. We describe an exploratory method that may be complementary to the aforementioned a priori counting methods, addressing some of the issues observed in the national cross-sectional studies.

#### 3.2.1. Description of Disability-Related Service Implementation and Data Collection in the PHC Setting

Since 1994, the nonprofit organization Tapori Paladines de la Felicidad has been offering comprehensive PHC for local villagers who live in the suburban north of Quito, Ecuador. Annually, around one thousand families, who mainly live within a radius of 10 km, make use of the services of the community Health Center. In 2000, an estimate of 39,619 people lived in that rural area. The population on the outskirts of the large city grew much faster than the national growth rate, and in 2020 almost 73,472 people lived in the same area, now suburban, with the population nearly doubling in 20 years. The distribution of age and ethnic groups is in accordance with national data and great national inequality also is reflected in the local population. In terms of income inequality, the families who use the PHC services mainly belong to the two poorest quintiles of the Ecuadorian and the local population. Every year, between 2500 and 3000 medical consultations are attended, for an average of 1200 different persons. Around 150 of them are persons with disabilities, who are seen in almost 20% of the total number of medical consultations.

Initially, services included health promotion, disease prevention, and curative care. In response to population demands, rehabilitation services were progressively integrated into the setting since 1996. At the time, there were no accessible health services designed to restore or improve the individual’s level of functioning in the village and surroundings. Persons with disabilities hardly participated in social life, and local conceptions of disability alternated between considering it either a punishment or a gift from God, to be accepted with resignation. Health professionals were confronted with persons with disabilities whose health-care-related needs were not addressed, despite options for services existed in theory. This marked the starting point of the gradual development of a community-based rehabilitation component in the Health Center.

Therapy initially focused on the development of motor and communications skills, based on a biomedical and individual disability paradigm. Through word-of-mouth advertising, persons with other functional difficulties found their way to the Health Center and implicit needs of persons with disabilities gradually converted into expressed demands. Daily experience taught that disability is far from being solely a biomedical health issue and the rehabilitation team, which initially consisted of a general practitioner, a psychologist, and a physiotherapist, grew into an interdisciplinary 14-member team that now offers, in addition to individual therapy, three modalities of day-care, learning, and working. Team members now opt explicitly for a biopsychosocial disability paradigm that takes into account all the components and interactions of the ICF.

The state-of-the-art of this evolving disability concept is reflected in interdisciplinary and multisector teamwork and in the nature and implementation of resources, for example, in the structure of the internal information system. Today, this two-part information system is structured as follows: a person-based archive with individual medical records of slightly fewer than 10,000 persons, and an administrative system that, in addition to logistical, financial, and organizational information, contains an Excel file (beginning in 2016) with consolidated disability-related data of all the persons whose individual medical record includes a specific file on functioning and disability. Since 2007, this individual specific file has been added to the medical file of all persons with disabilities who have ever used the services of the Health Center from 1994 to the present day. This is done gradually for new persons with disabilities or retroactively for persons who have previously used the service. In the consolidating administrative Excel file, the person with disability is included only once and is never excluded, regardless of whether the use of services is constant or not, or has been discontinued in the meantime, by death or for any other reason. There are no multiple entries of the same service user over time.

The team’s GPs add the specific file to the individual health record when individuals, their close relatives, or, ultimately, team members of the Health Center perceive and report impairment or biological dysfunction, lasting difficulties in performing daily life activities, and participation in society. Disabling health conditions and impaired body structures and functions often trigger visits to the Health Center. Nonetheless, disability identity, outsourced by the use of the specific file that includes the ICF diagram, is only assigned when difficulties exist in all three ICF functioning domains, including activity limitations and restrictions in participation, and when the person cannot fulfill the social role expected by the local population [29]. The use of this local population’s norm of functioning provides simultaneously information about environmental barriers and facilitators.

No core questions about specific functional difficulties are used in the Health Center to determine whether someone is or is not a person with disability and eligible for services, and no thresholds on the scale of human functioning other than those that are socio-culturally determined are applied. Yet, daily practice has also shown that strictly dividing the population into “persons with disabilities” and “temporally nondisabled persons” is difficult [30]. For this purpose, an internal agreement applies that data on a person with disability are only included in the consolidating Excel file when the information from the individual specific disability file, summarized in the ICF diagram, has been validated by the individual or, by default, by a family member or guardian, and by consensus of the rehabilitation team.

In the summarizing administrative Excel file, the research team categorizes available and useful data from the individual files according to the ICF classification, including specific data, such as the age of mothers at delivery in the case of disabling prenatal or perinatal biomedical and contextual factors, or data on specific important life events, corresponding to the personal factors component of the ICF. The information in the consolidated Excel file is updated quarterly with data on new admissions or on pertinent changes in a person’s disability situation. New information is added without deleting previous data. Data are also not deleted when, for example, someone stops using the service or in the event of death. In this way, every person with disability who makes use of the services in the Health Center has an individual report on disability-related events and functioning during the life course [31]. The consolidating Excel file brings together those histories and can be a valuable input for a local population’s disability profile.

In a previous paper, we described in detail the process of implementing the ICF in the individual medical files of persons with disability in the PHC setting, and we present the case of a person with disability during life course with six connected ICF diagrams [29]. In the current paper, we focus on how the universe of persons with disabilities is defined in the PHC setting, including them in the consolidating administrative Excel file. This allows for calculating disability prevalence and enables the analysis of disaggregated data. Future papers, for example, on biomedical and contextual disabling factors and their relationship, will explain and discuss in detail how functioning data are summarized in the Excel file, again mainly based on the ICF framework.

#### 3.2.2. Disability Period Prevalence in the Health Center

A priori identifying and counting of persons with disabilities was not a goal in this PHC setting, and it was certainly not a prerequisite for proffering services. However, as a collateral result of the dialectical process of identifying needs and service implementation, the PHC setting has a historical archive with validated information on persons with disabilities. These data make it possible to calculate the period prevalence of disability from the general population sample, understood as the ratio of the number of persons with disabilities to the number of all the persons who have used the Health Center’s services from 1994 until 2020.

During this period, we opened 10,106 general clinical files and disability identity was assigned in 473 cases (See Figure 2). Longitudinal disability counting over twenty-seven years in the Health Center yields a rate of 4.68%, a value within the range of the self-reported prevalences in the censuses, but considerably lower than the rate from the WHS and the estimated global prevalence of 15%.

The list includes persons with disabilities at all stages, children as well as adults, and persons with all types of impairment. Almost half of this group experiences severe functional difficulties, defined as a condition in which the support or care of a third person is required, most of the time or permanently. This results in an estimated rate of persons with severe disabilities that is close to the average prevalence of persons with significant difficulties of 2.2% in the WHS [4]. The remaining persons with disabilities, almost 2.5% of the whole population sample, present mild to moderate limitations, sometimes subtle and difficult to observe at first sight, but always with such an impact on daily life so that services beyond the normally expected are required. This estimated prevalence of persons with mild to moderate disabilities is much lower than the estimation of the WHS, which indicates that around 13% of the world’s population experiences mild to moderate disability. Other data disaggregation on age, gender, biomedical etiology of dysfunction, disability type, some social and environmental barriers and facilitators, among others, is possible but beyond the scope of this paper.

#### 3.2.3. Reliability and Validity of Period Prevalence

Researchers’ bias related to the biomedical entry-point for identification of persons with disabilities could be a reliability concern. Nonetheless, we consider that the risk is minimal because the health professionals imposed no predetermined or fixed definition of disability. In addition, over time, the concept of disability evolved from a biomedical paradigm focused on impairment to a broader biopsychosocial disability paradigm that links disability with social and environmental barriers, and difficulties with the full exercise of human rights. Disability identity is never imposed but is mainly assigned based on self-perception of difficulties with functioning by the person with disability, or, by default, based on proxy reporting by close relatives or health professionals, and is supported by validated information in the ICF diagram.

By not using objective indicators or a validated measurement scale, the definition of disability and the assignment of the identity of a disability can be perceived as subjective and affect credibility of the findings. Yet, continuity of care in this locally embedded PHC setting enhances the internal validity of disability rating as the observational and participatory collection of data over time allows for precise and updated information, and minimizes false positive or negative results. Well-functioning people will not persist in presenting themselves as a person with disability for benefits that are obtained only by working to improve functioning and by investing personal resources. Moreover, trust and a long-term relationship decrease shame and perceived stigma. Continuity of care also favors the identification of persons with mild disabilities whose functional problems may be less noticeable. Their risk of exclusion from eligibility for services is minimal because no a priori criteria or a posteriori cutoff related to disability types or severity of functional difficulties are applied.

Regarding external validity, the PHC disability rate cannot be generalized. The Health Center’s universe of patients is not a representative sample of the total Ecuadorian population and not even for the total local population, especially in terms of economic inequality and geographical diversity. Otherwise, the implementation of disability-related services probably attracted persons with disabilities for whom services were not available or accessible before.

## 4. Discussion

The discussion on variations in disability rates affirms that disability counting is not an objective and uniform procedure, focusing the attention on influencing factors at risk of being excluded from services in the a priori identification of persons with disabilities. We suggest considering contributions of longitudinal needs-driven a posteriori disability counting for local policy planning and service implementation, complementary to cross-sectional disability counting, and we present a reflection on the use of disability rates for the evaluation of disability situations from the perspective of PHC in the Global South.

### 4.1. Disability Counting in Search for Balance Between Those Who Count and Those Who Are Counted

Our analysis of reasons for variations in national disability rates corroborates the “illusory character of the disability concept as an entity of known and fixed dimensions” [2]. Before using data for policy planning, service implementation, and evaluation, we must keep in mind the characteristics and shortcomings related to the design of any disability counting study as data differ because “operational definitions and measures of disability vary widely according to the purpose for which the data are collected, the concept of disability used and the data collection methods and instruments applied” [32]. We add reliability issues to this list of points of attention as our analysis of two cross-sectional studies, whose results were taken for granted by the Ecuadorian government and by international organizations, unfortunately affirms the WHO’s statement about inaccuracy of data.

In line with current international recommendations, we suggest joint use of different cross-sectional methods to overcome disadvantages and shortcomings of each, as separate use is not sufficiently accurate to determine the function of populations [3,11,33]. The implementation of a short set of questions on functioning, for example, the Washington Group Short Set, can be used during censuses as screening device to identify a sample for a postcensus survey, for example, the Model Disability Survey of WHO [34,35]. In this survey, sufficient negative responders in censuses should be included in order to still identify persons with disabilities and to take them into account for service planning. Data can also be crossed with one-time administrative data collections. This can only be done in a coordinated way and when data are relatable [3]. The ICF already provides a universal conceptual framework for this.

Yet, a still worrying observation is that all of the cross-sectional identification methods first define who belongs to the group of persons with disabilities and then decide on the services to be implemented, based on characteristics and needs of this established separate universe. These a priori counting methods entail the risk of leaving persons with disabilities behind as decisions on disability paradigm, methods for data collection, and thresholds may already exclude them from this universe at the outset. This turns politicians, members of the international community, researchers, epidemiologists, and health professionals into powerful intervening actors, especially when decisions are not always made on a scientific basis, but respond to practical, ideological, and even political issues. Data from the national studies show that children, persons with intellectual disabilities, and persons with mild and moderate disabilities are especially at risk. Yet, their a priori exclusion from services misses the opportunity to prevent their probable deterioration due to unmet disability-related needs, while focusing on intervention in these groups would likely produce the most gain in population health [6,8]. *Those who count* must keep in mind that they bear responsibility in determining whether individuals are entitled to services or left behind with unmet disability-related needs. In order to not only generate data and measure exclusion in the predefined group of persons with disabilities but to also identify and include a priori excluded persons, the cross-sectional methods can be combined with longitudinal counting methods, for example, continuous administrative registration of data as part of the normal operation of a service or program or the PHC method presented.

From the side of *those who are counted*, our findings on underreporting and the resulting exclusion from services are inconclusive. Persons may not perceive their disability as they have interiorized their difficulties as normal, or they may not know about the options for dealing with difficulties, as was seen when starting up the service in the PHC setting. Otherwise, it might be interesting to consider social resilience, understood as the ability of social actors to cope with adversities, as an intervening factor when exploring why the percentage of persons with mild to moderate disabilities is much lower in the PHC setting in a developing country than in the WHS, while both use a biopsychosocial disability paradigm. Without denying the negative effect of stigmatization and negative cultural attitude, we suggest further research into generally accepted statements regarding underreporting, since a negative answer on a closed question about disability identity during censuses is perhaps not always a culturally negatively influenced answer but rather a reflection of functioning coping mechanisms at the individual and social level.

### 4.2. A Contribution to Disability Counting As a Collateral Result of Meeting Needs in Primary Health Care

The *upside–down* method for disability counting in the PHC setting, where disability-related health services were gradually implemented based on perceived needs and demands without any a priori searching for and counting of persons with disabilities, is worthy of further investigation, based on the arguments that support reliability and validity of the estimated disability rate. The aforementioned problem, that knowing how many persons have disabilities is not the same as knowing their needs, is not an issue here. First, needs are identified based on self-perception, without thresholds on the continuum of functioning. Only then persons will be assigned to the universe of persons with disabilities, which provides data on disability to the consolidating information system. Continuity of care strengthens the identification of persons with disabilities, which are not as readily apparent as physical or sensory disabilities, for example, children with behavioral problems or with learning difficulties, and persons with mild difficulties or with mental problems. The identification of these persons, who perhaps were not taken into account in cross-sectional counting, may be a valuable contribution of disability-inclusive PHC to the reduction of the risk of their exclusion from services and to the development of an accurate local population’s disability profile, as disability data on national level may not reflect intracountry differences or specific characteristics of which being aware of is important for local disability policy planning and service implementation [5].

These findings prompt to check reliability and validity of disaggregated data from the consolidating Excel file and to explore their usefulness, not only for the persons with disabilities but also for the temporally nondisabled persons. For example, a prevention program that tackles specific social characteristics can be developed based on the analysis of local data on biomedical etiology of disabilities. Disaggregated local data can also be compared to national data to learn from differences, in particular on disability types or age groups that may be over- or underreported.

In response to the generalizability concern, we suggest a pilot study on the adoption of the Health Center’s method for identifying and addressing disability-related needs in other PHC settings as universal coverage in accessible Health Centers and the joint use of data from different sources may favor internal and external validity and contribute to the evaluation of intracountry differences. Similar to the sentinel general practices in Europe that help policy makers define and assess health and disability policies, data from their registers of service utilization could be combined and cross-analyzed [36]. By upscaling IT systems, data could also be triangulated with data from cross-sectional initiatives such as population censuses or, as suggested by WHO, with information from social registration systems on persons with disabilities who receive benefits based on a programmatic need [3].

### 4.3. Comparing Rates to Evaluate Disability Situations and Policy Implementation May Be a Walk on a Tightrope

We present this reflection on the comparison of disability rates to evaluate disability situations, as thinking about disability counting from the perspective of PHC in the Global South led to the questioning of some fixities [37]. There is no doubt that standalone disability rates may be useful for raising awareness, planning services, and assigning resources. However, rates are also used to evaluate disability situations and implement policies. The evaluation process implies comparison and thus raises questions about which indicators are to be compared, and to what values.

One problem with disability rates is that they only reflect part of the complex reality of human functioning. In addition to comparing numbers of persons with disabilities, differences in the range of disability types and the degree of functional limitations should be analyzed. We must also take into account disabling factors and unmet needs, existing services and service utilization, and even different perceptions related to the same “objective” difficulties with body structures and functioning. When using data for evaluating disability situations and policy implementation, disability statistics should look beyond the scope of disability rates and not be limited to the comparison of numbers of persons with disabilities.

A second pitfall when using disability rates for evaluation relates to which specific values are being compared. In the case of this study, prevalences from different studies carried out during particular periods within a population group are compared. Bearing in mind the aforementioned research-design-related reasons for differences in cross-sectional studies, rates should be checked for their mutual comparability before drawing any conclusion from the differences. Disability rates from different population groups are also often compared with the average of many other rates, such as the comparison of national rates with the global prevalence of 15%, or as relative values among each other.

Yet, the comparison of different relative values is not that simple and there is great temptation to consider one value as an absolute and correct reference or golden standard, as, for example, in the case of cross-country comparisons. In this comparison, it is expected that low- and middle-income countries will yield higher disability rates than countries with high incomes, as it is generally assumed that persons with disabilities are more likely to be poor and that poverty contributes to perpetuating disability [32,38]. Low-income countries, however, consistently report lower prevalence rates and “observed differences whereby the most developed countries show the highest disability rates and the least developed countries the lowest are counterintuitive” [39]. Reasons for this unexpected result are generally presumed to relate to underreporting in low- and middle-income countries. The rates of high-income countries are hardly questioned and taken for granted as reference values.

Nevertheless, a slippery slope of flawed arguments may lead to false conclusions because “longitudinal data sets to establish the causal relation between disability and poverty are seldom available, even in developed countries” [4], and the correlation between disability and poverty might be more complicated than a simple disability–poverty cycle [32]. Extrapolating from the WHO statement that just as “a gold standard for which the threshold line should be drawn on the individual continuum of functioning does not exist” [4], there is no absolute reference point or gold standard for the level of functioning in population groups.

From the perspective of cultural relativism, instead of debating who is wrong or right when using disability rates to evaluate disability situations and policy implementation, it might be better to focus on why differences exist and what they tell us. A fair reference point might be individual or local populations’ expectations about functioning. If so, longitudinal self-reporting without the implementation of a priori criteria or any a posteriori cutoff, such as used by the PHC setting, would be a legitimate approach to counting disability.

## 5. Implications of Findings

Identifying and counting persons with disabilities, expressed in terms of national disability prevalence, yields confusing information. Statistics should be handled with care when it comes to local disability policy planning, service implementation, and evaluation. They carry an insidious risk of a priori exclusion of persons with disabilities from services because of researchers’ methodological decisions, as the analysis of the six cross-sectional Ecuadorian national studies has shown. Those who are not counted remain invisible for disability policy.

A needs-driven method for identifying persons with disabilities and collecting disability data in a community-based PHC setting can contribute to local disability policy planning, by providing more nuanced information about a local population’s disability profile and reducing the risk of exclusion from services. Counting disability in this way reflects an evolving disability paradigm and moves beyond a quantitative a priori and dichotomous approach. It also shows the importance of disability counting from a personal perspective, understanding disability as a universal experience, without thresholds on the continuum of functioning, and from a life course approach. We suggest that data from different PHC’s can be gathered in a systematic way and be combined to complete cross-sectional dataset. A comparison with national and intranational datasets can lead to interesting analyses, for example, of within-country and cross-country variations.

Our reflections on the comparison of disability rates to evaluate disability situations and policy implementation invite us to think about why differences exist and what these differences tell us, rather than discussing who is wrong or right. We suggest future research on statements about underreporting, where so far only stigmatization and negative cultural attitude are pointed out as causes.

Our findings are in line with current international guidelines for disability measurement and statements on the need to combine different disability measurement methods in order to overcome disparities and confusing data and to avoid exclusion. From a disability-inclusive PHC perspective, we share an a posteriori longitudinal data collection experience, mainly based on self-reported functional limitations and assessed with the ICF, which can be combined with a priori counting methods, objective impairment screeners, and other rapid assessment tools for activity limitations and participation restrictions. Our findings encourage building the capacity of local stakeholders to collect and analyze disability data and to contribute to the development of comprehensive modular tools in order to broaden and refine disability data for disability policy.

## 6. Study Limitations

A possible weakness in disability counting in the PHC setting is the perception of subjectivity in defining who is and who is not a person with disability, as we did not use any objective instrument for measuring disability, except for an open ICF-based framework. A second weakness is that only local data are provided, which might lead to selection bias and not be representative for the total population. This could be overcome by organizing “sentinel PHCs” on a national scale. We emphasize that this study is exploratory and will contribute to further research on our findings and ideas, exploring validity and usefulness of disaggregated data for local population’s disability planning. The usefulness of health service utilization as proxy-for-need in disability counting may be questioned and limited, as there is strong evidence that persons with disabilities have poor access to health care [33]. In theory, PHC strives for universal coverage and provides services from a life course perspective. Yet, questioning both demand-side and supply-side determinants of PHC accessibility in the field is justified and the search for evidence on approachability, availability, acceptability, appropriateness, and affordability of services and care, and how to face difficulties, gaps, and barriers emerges as yet another topic for future research.

## 7. Conclusions

The analysis and comparison of disability rates from the national cross-sectional studies confirm that dichotomous counting is influenced by the researchers’ methodological decisions and yields data that are confusing and may not be useful for local disability planning. Data collection on disability in a community-based PHC setting, without any a priori searching for and counting of persons with disabilities but along with the dialectical process of identification of needs and service implementation, may provide a reliable and credible estimate of disability rate and other data for local disability planning and service implementation. Given the exploratory character of our study, findings on the a posteriori method for disability counting provide ideas and insights that may improve matching disability-related needs with local health service delivery, and reduce the risk of exclusion from services. It may open new paths for future research on its complementarity with a priori cross-sectional methods for counting persons with disabilities, as a contribution to more nuanced data for local disability policy and to the global change in disability thinking and counting.

## Figures and Tables

**Figure 1 ijerph-18-05103-f001:**
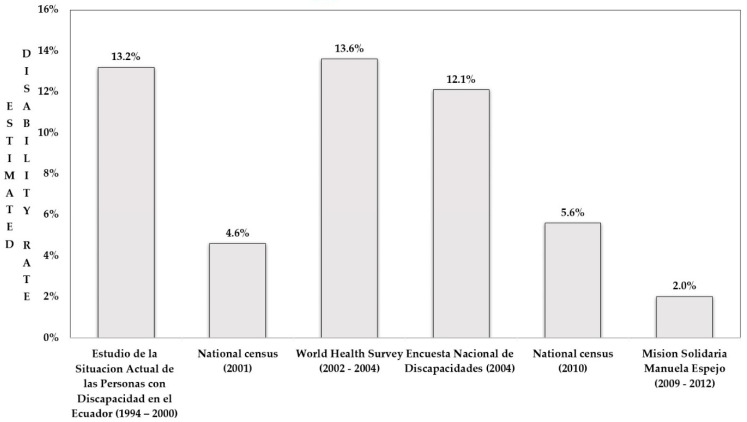
Estimated disability rates from six cross-sectional Ecuadorian studies.

**Figure 2 ijerph-18-05103-f002:**
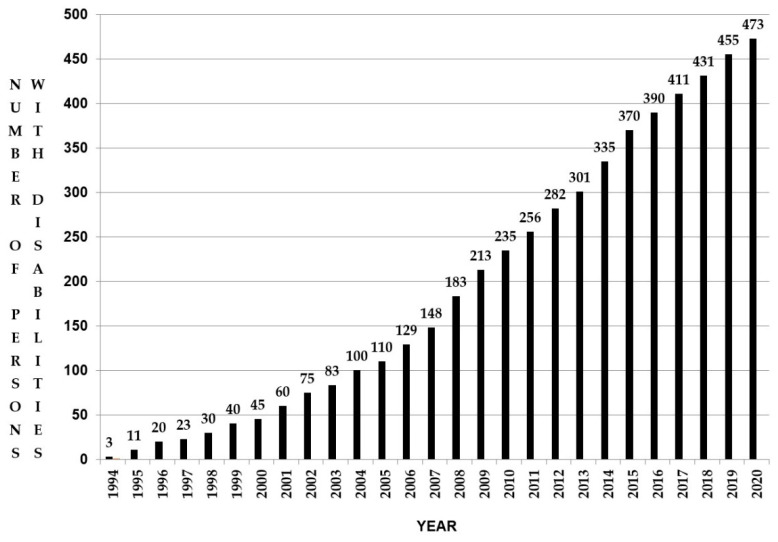
Cumulative number of persons with disabilities in the Tapori primary health care setting.

**Table 1 ijerph-18-05103-t001:** Summary of selected characteristics of the six national cross-sectional studies on disability data (1994–2012).

Study	Purpose of Disability Counting	Disability Paradigm	Age	Sampling Method and Sample Size	Data Collection Method	Result (Disability Rate)
Estudio de la Situación Actual de las Personas con Discapacidad en Ecuador (ESADE) 1994–2000 [15]	Descriptive study about the situation of persons with disabilities, in order to identify needs and for service planning	Biomedical unilateral causation; No information on thresholds	Older than 5 years	Stratified sampling at national level, survey of 68,000 persons in 16,800 households	General questionnaire on deficiencies (head of the household) + Specific questionnaire for the identified persons with impairments, equated with disability type(s)	13.2% Approximately 1,600,000 persons with disabilities
VI National Census in 2001 [16] VII National Census in 2010 [14]	A priori disability screener as starting point for national disability policy, for in-depth survey sampling, and for evaluation of disability situation at national and aggregate level	No initial disability definition; After initial disability identification: biomedical focus on disability types; No thresholds	All ages	General population sample at national level 2001: 12,156,608 persons 2010: 14,483,499 persons	Self-reporting by answering “yes” or “no” to the question “Do you present permanent disability for longer than a year?” If yes—answer: questions on type of impairment	2001: 565,560 persons = 4.6% 2010: 816,156 persons = 5.63%
World Health Survey (WHS) 2002–2004 [17]	Monitoring and evaluating national disability policy and cross-country comparison of data	Components of health classification, synchronized with the ICF; Use of degree of functioning difficulties and “a posteriori” cutoff	18 years and older	Stratified sampling at national level, survey in 6135 households	Open questioning in face-to-face household survey, with questionnaire on 8 health and functioning core domains	13.6%
Encuesta Nacional de Discapacidades (END) 2004 [18]	To complement the information of the ESADE study with the social vision of the ICF and to meet the need for universal language for comparison at aggregate level	Biopsychosocial disability paradigm (ICF) “a posteriori” division in mild to moderate and severe limitations	All ages	Random sampling at national level, using the INEC national sampling frame, survey of 83,043 persons in 19,608 households	Open questioning in a base survey and in-depth module for persons with impairment, activity limitations and participation restrictions	12.14% 1,608,334 persons
Misión Solidaria Manuela Espejo (MSME) 2009–2012 [19]	Action research for immediate service implementation in response to critical cases	No initial disability definition; Biomedical “defectology” focus; Only severe functioning limitations were included	All ages	General population sample at national level, with 1,167,893 households visited	Self-reporting in response to a national call for disability identification + Health and impairment-based survey with closed questions	2.02% 293,743 persons

**Table 2 ijerph-18-05103-t002:** Distribution of deficiencies that originate disability in persons older than five years (data are from Reference [18]).

Deficiency Type	Frequency of Deficiencies	Percentages
Hearing	73,600	4.6%
Speech and language	113,600	7.1%
Visual	80,000	5%
Musculoskeletal	144,000	9%
Visceral	83,200	5.2%
Disfiguring	49,600	3.1%
Intellectual	432,000	27%
Psychological	624,000	39%
TOTAL	1,600,000	100%

**Table 3 ijerph-18-05103-t003:** Persons with disabilities, grouped by age and by impairment and degree of activity limitation and participation restriction (data are from Reference [16]).

Persons with Disabilities	Number	Percentages
Persons with disabilities, younger than five, who present activity limitation and participation restriction	17,838	1.11%
Persons with disabilities, five years and older, who present impairment	702,793	43.7%
Persons with disabilities, five years and older, who present mild to moderate limitation	247,520	15.39%
Persons with disabilities, five years and older, who present severe limitation	640,183	39.8%
TOTAL	1,608,334	100%

## Data Availability

Data presented in this study are available on request from the corresponding author. The data from the administrative Excel file are not publicly available due to information that could compromise the privacy of research participants.

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
