# Peer review of "Lessons from Disability Counting in Ecuador, with a Contribution from Primary Health Care"

_ijerph, 2021, doi:10.3390/ijerph18105103_

Round 1

Reviewer 1 Report

I consider that the problem of the article could be an adequate survey of health policies related disability, but the way the topic is exposed does not allow for a rigorous scientific analysis, nor does it show the scientific evidence of the study. The manuscript is presented in accordance with the guidelines of the IJERPH. It presents importante scientific weaknesses, namely related to the definition of the research objectives and methodology, that compromise the entire presentation of the article.

Reviewer 2 Report

In my opinion, this paper provides a very innovative and suggestive insight on the complex problem of disability measurement. I think that the methodology of the research is very appropriate, and the discussion and the conclusions are supported by the results of the research. The contributions of the paper merit to be discussed and may open new paths for future research.

I have only two general remarks that I would like the authors to discuss if possible.

1) If I have not misunderstood the position of the authors, it seems that they advocate for a method of counting disability that relies mainly on self-perception of disability, and in lines 421-422 they write that “a no-answer on a closed question about disability  identity during censuses is perhaps not always a culturally negatively influenced answer but the reflection of well-functioning coping mechanisms and resilience”. This point of view seems to me very interesting, and I think it is very important to consider the opinion of those who are counted. But I think the authors underestimate a vey important factor that can lead to a low self-perception of disability: the lack of empowerment of persons with disabilities. Many persons with disabilities do not have the perception of their disability because they are used to a deficit of activity or participation that they have interiorized as normal, and have not been made aware of their rights and of the possibility of overcoming those deficits. Therefore, I think that an accurate measurement of disability should combine the attention to self-perception of disability with more objective indicators.

2) The second remark is very closely related to the precedent one. I suggest the authors to pay attention and to include in the article a reference to the definition of disability in the Convention on the Rights of Persons with Disabilities (Preamble and Article 1), which links disability with the barriers to social participation and full exercise of human rights. This should be, in my opinion, the main indicator to identify and count persons with disabilities, although it seems clear that barriers are very difficult to measure. I think an explicit reference to this concept and to the social model of disability that underlies the Convention would improve the article.

Finally, I think the authors overestimate the influence of those who count disability, suggesting, although they not say it explicitly, that the method of counting disability may be based mainly on ideological or political reason (see above all page 9 of the paper). In my opinion, the different methods of disability counting are rather based on scientific paradigms.

These are my views, that I put forward only with the purpose of improving an article which seems to me very interesting and of a very high quality.

Moreover, although English language and style is not bad, I suggest the authors to do a revision of the text, because some of the constructions are not fully elegant.

Author Response

Please see the attachtment.

Reviewer 3 Report

This paper from a LMIC country is a welcome addition to the international literature and would be well placed within the new disabilities section of the journal.  It deals with the thorny issue of how disability is defined within administrative datasets and presents comparative data from various surveys and censuses undertaken in Equador over the past two decades.  Moreover, the paper presents a small-scale example as to how primary healthcare services could contribute to identification of the persons with disability and their needs. 

I have a number of suggestions which the authors might consider when revising the paper.

For an international readership it would be useful to have some information about Ecuador: size of population; age distribution, life expectancy, socio-economic indicators (including poverty) and ethnic/cultural mix. 

It might be better to present the paper in two parts. Part 1 dealing with past surveys and Part 2: with the PHC data.  They are qualitatively very different.

Part 1 is worthy of presentation in its own right.  Here it would be helpful to the readers if the authors created table that summarised the characteristics of the various national data sources they reviewed: Sample size; sampling method; how information was obtained; how disability was identified; percentage with disability.  The text could then elaborate on the differences across the surveys.  Figure 1 would not then be necessary (although I appreciate the visual impact it presents.)

I wondered too if the authors needed to give more attention to any further disaggregation of the disability data that is available from the data sources: for example by age - children: working age adults and elderly persons – but also by gender and levels of poverty.  One overall figure could mask important sub-group variations

In the discussion the authors also need to refer to the possible impact of different methods for collecting data – self-completion of forms; interviews etc.

Part 1 could usefully end with a summary of the deficiencies in current information gathering to set the scene for Part 2: ‘The Case of Tapori ‘ which hopefully will illustrate how PHC data can address some of the shortcomings.

More detailed is required about how the information is gathered and stored on the PHC system and any validity checks that are made.  It would also be helpful to know more about the characteristics of the area served by the PHC service (perhaps as variations from the country characteristics reported in Part 1).  

The authors need to expand on the all-too brief account they provide as to the criteria used to determine  persons with disability  (“[through]the dialectical process of identification of needs and service implementation, we can calculate the period prevalence of disability” line 257/58).  Also the criteria used for ‘severe functional disabilities (line 264).

The discussion in this part could focus on how the PHC systems can be scaled up nationally in order to account for the intra-country (that is regional variations) in disability reporting that seem to be present in all nations.  How prepared are the current IT systems in Ecuador to take on the role of pooling “data from their registers of service utilization could be combined and cross-analyzed” (line 362/3)?

The final, summarising discussion could be reduced if some of the present content is moved to the two Parts as proposed above.  Also I found the section from lines 380 onwards less connected to the core arguments that had been made previously as it opens up a further debate about policy implementation and evaluation that goes beyond the scope of this paper.   

Some more minor points.

The title could be revised: Lessons from disability counting in Ecuador: The contribution of Primary Health Care.  The abstract might also be adjusted to reflect to dual aspect of this paper.

Line 76: The term ‘within-country differences’ is ambiguous as it is also applied to differences in disability prevalence across regions within a country.  It is sufficient to state: large variations in disability prevalence in Ecuador.  (The issue of intra-country (inter-regional) variation in important in relation to the argument for PHC systems).

Regarding the statement on ethical approval, did this cover particularly the data from PHC in Tapori?  If so this should be clearly stated as I would not have thought ethical approval was required to undertake the comparisons made across national surveys reported in Part 1.

The references are not numbered in the style used by the journal.  

Reviewer 4 Report

According to the authors “This paper aims to contribute to the debate on disability counting, moving away from dichotomous snapshots 9 towards a longitudinal way of counting on a continuum of human functioning”,

however, the objective that this study is trying to achieve is not very clear.

In the introduction section, the justification of the study supported by bibliographic reviews on the subject or relevant studies should be developed a little more.

Do the authors carry out a biliographic review of the subject under study? In methodology it is mentioned that the authors examined printed and electronic documents focusing on specific topics such as the definition of disability, but How many documents besides the six cross-sectional research projects on disability? How did you proceed to retrieve those documents? Who located them?

Later in results they say that they analyze of six cross-sectional studies and one longitudinal research project in Ecuador

The MATERIALS AND METHODS  section should be further developed to understand how the data collection has been carried out.

in the conclusión section the authors give a personal opinion on the subject, content that should be included in the discussion section.

Round 2

Reviewer 1 Report

The authors made a thorought revision of the device, considerably improving the weak points of the first version, as were the definition of the research objetives, the material and methods and the presentation of the results and discussion. There are small rooks that must still be reviewed, namely the implications that the study has for the science and reviewing the references, some are incomplete (ex: reference 29).

Author Response

Quito, April 10, 2021

To,

The Editors,

Dr. Manjula Marella

Dr. Islay Mactaggart

Guest Editors

Dear Section Managing Editor

Miya Zhang

We want to thank you and the reviewers for allowing us to submit a revised version of our manuscript ‘Lessons from Disability Counting in Ecuador, with a Contribution of Primary Health Care’, for consideration in the Special Issue “Measuring Disability and Disability Inclusive Development” of International Journal of Environmental Research and Public Health.

We thank the reviewers for their almost complete approval of the previous manuscript. In answer to reviewer 1´s observation, we redrafted the conclusion and we specified implications of our study for science and study limitations (Lines 458 – 495). The English language and style have been checked and we reviewed the references. Observations and comments are welcome.

With the consent of my co-authors, Pedro Celestino Álvarez Vera, Ximena del Carmen Pavón Benítez, Celia Katherine Rosero Arboleda, Peter Prinzie and Jo Lebeer, I hereby send you the revised manuscript as requested. We hope our revision meets the wishes and concerns of both you and the reviewers.

Looking forward to hearing from you,

Sincerely,

Inge Debrouwere.

Response to reviewer 1: Please see the attachment.

Reviewer 3 Report

Many thanks for the fulsome consideration you have given to my suggestions.  Congratulations on conducting such a thorough study of existing surveys and for creating a recording system for use in PHC that will facilitate more accurate representations of persons with disabilities and provide a basis for individualised planning to meet their needs.   I hope other PHC districts in Ecuador will adopt this methodology.  You also have much to teach the wider international community. 

Author Response

Quito, April 10, 2021

To,

The Editors,

Dr. Manjula Marella

Dr. Islay Mactaggart

Guest Editors

Dear Section Managing Editor

Miya Zhang

We want to thank you and the reviewers for allowing us to submit a revised version of our manuscript ‘Lessons from Disability Counting in Ecuador, with a Contribution of Primary Health Care’, for consideration in the Special Issue “Measuring Disability and Disability Inclusive Development” of International Journal of Environmental Research and Public Health.

We thank the reviewers for their almost complete approval of the previous manuscript. In answer to reviewer 1´s observation, we redrafted the conclusion and we specified implications of our study for science and study limitations (Lines 458 – 495). The English language and style have been checked and we reviewed the references. Observations and comments are welcome.

With the consent of my co-authors, Pedro Celestino Álvarez Vera, Ximena del Carmen Pavón Benítez, Celia Katherine Rosero Arboleda, Peter Prinzie and Jo Lebeer, I hereby send you the revised manuscript as requested. We hope our revision meets the wishes and concerns of both you and the reviewers.

Looking forward to hearing from you,

Sincerely,

Inge Debrouwere.

Response to reviewer 3: Please see the attachment.

Reviewer 4 Report

The authors have made all the changes indicated.

Author Response

Quito, April 10, 2021

To,

The Editors,

Dr. Manjula Marella

Dr. Islay Mactaggart

Guest Editors

Dear Section Managing Editor

Miya Zhang

We want to thank you and the reviewers for allowing us to submit a revised version of our manuscript ‘Lessons from Disability Counting in Ecuador, with a Contribution of Primary Health Care’, for consideration in the Special Issue “Measuring Disability and Disability Inclusive Development” of International Journal of Environmental Research and Public Health.

We thank the reviewers for their almost complete approval of the previous manuscript. In answer to reviewer 1´s observation, we redrafted the conclusion and we specified implications of our study for science and study limitations (Lines 458 – 495). The English language and style have been checked and we reviewed the references. Observations and comments are welcome.

With the consent of my co-authors, Pedro Celestino Álvarez Vera, Ximena del Carmen Pavón Benítez, Celia Katherine Rosero Arboleda, Peter Prinzie and Jo Lebeer, I hereby send you the revised manuscript as requested. We hope our revision meets the wishes and concerns of both you and the reviewers.

Looking forward to hearing from you,

Sincerely,

Inge Debrouwere.

Response to reviewer 4: Please see the attachment.
